# IUSMMT: Survival mediation analysis of gene expression with multiple DNA methylation exposures and its application to cancers of TCGA

**Zhonghe Shao**[1‡], **Ting Wang**[1‡], **Meng Zhang**[1], **Zhou Jiang**[1], **Shuiping Huang**[1,2,3]*, **Ping Zeng**[1,2,3]*

**1** Department of Biostatistics, School of Public Health, Xuzhou Medical University, Xuzhou, Jiangsu, China, **2** Center for Medical Statistics and Data Analysis, Xuzhou Medical University, Xuzhou, Jiangsu, China, **3** Key Laboratory of Human Genetics and Environmental Medicine, Xuzhou Medical University, Xuzhou, Jiangsu, China

‡ These authors are co-first authors on this work.
* hsp@xzhmu.edu.cn (SH); zpstat@xzhmu.edu.cn (PZ)

**Data Availability Statement:** All files are available from the TCGA database (https://portal.gdc.cancer.gov/legacy-archive/).

## Abstract

Effective and powerful survival mediation models are currently lacking. To partly fill such knowledge gap, we particularly focus on the mediation analysis that includes multiple DNA methylations acting as exposures, one gene expression as the mediator and one survival time as the outcome. We proposed IUSMMT (intersection-union survival mixture-adjusted mediation test) to effectively examine the existence of mediation effect by fitting an empirical three-component mixture null distribution. With extensive simulation studies, we demonstrated the advantage of IUSMMT over existing methods. We applied IUSMMT to ten TCGA cancers and identified multiple genes that exhibited mediating effects. We further revealed that most of the identified regions, in which genes behaved as active mediators, were cancer type-specific and exhibited a full mediation from DNA methylation CpG sites to the survival risk of various types of cancers. Overall, IUSMMT represents an effective and powerful alternative for survival mediation analysis; our results also provide new insights into the functional role of DNA methylation and gene expression in cancer progression/prognosis and demonstrate potential therapeutic targets for future clinical practice.

## Author summary

DNA methylation has a causal effect on tumorigenesis and gene expression may be an important mediator of such influence. However, inferring the existence of mediation effect of gene expression is statistically challenging, especially when multiple and even high-dimensional DNA methylation exposures are collectively analyzed. To solve such challenge, we developed a new mediation approach called IUSMMT, in which mediation effects are determined by two separate tests: one for the association between methylations and the expression, the other for the association between the expression and the survival

**Funding:** The research of PZ was supported in part by the Youth Foundation of Humanity and Social Science funded by Ministry of Education of China (18YJC910002), the Natural Science Foundation of Jiangsu Province of China (BK20181472), the Chinese Postdoctoral Science Foundation (2018M630607 and 2019T120465), the QingLan Research Project of Jiangsu Province for Outstanding Young Teachers, the Six-Talent Peaks Project in Jiangsu Province of China (WSN-087), the Training Project for Youth Teams of Science and Technology Innovation at Xuzhou Medical University (TD202008), the Postdoctoral Science Foundation of Xuzhou Medical University, the National Natural Science Foundation of China (81402765), and the Statistical Science Research Project from National Bureau of Statistics of China (2014LY112). The research of SH was supported in part by the Social Development Project of Xuzhou City (KC19017). The research of TW was supported in part by the Social Development Project of Xuzhou City (KC20062).The funders had no role in study design, data collection and analysis, decision to publish, or preparation of the manuscript.

**Competing interests:** The authors have declared that no competing interests exist.

outcome conditional on methylations. IUSMMT effectively combines the evidence of the two tests and infers the emergence of mediation effect by fitting an empirical three-component mixture null distribution. To evaluate the performance of IUSMMT, we conducted extensive simulation and analyzed ten TCGA cancers. Overall, we demonstrated the robustness, validity, and utility of IUSMMT under a wide variety of scenarios.

This is a *PLOS Computational Biology* Methods paper.

## Introduction

Many recent studies have revealed that epigenetic abnormalities, particularly aberrant changes in methylation, exert an important causative effect on complex diseases [1,2]. However, the mechanisms regarding how alterations of DNA methylation influence diseases remain largely elusive. Biologically, it has been widely acknowledged that DNA methylation leads to a direct functional modification of the genome by regulating gene expression [3–13], which reversely affects various diseases [14–17]. Statistically, this motivates researchers to consider gene expression as a critical causal mediator of DNA methylation on the development of diseases. However, the principal genes that control the pathogenesis of diseases have not yet been identified and the underlying mechanisms of causal genes on diseases are also unclear.

With respect to statistics, mediation analysis, particularly popular in sociology, epidemiology, and psychology [18–22], offers a flexible means to interpret the interplay between DNA methylation, gene expression and diseases. The essential requirement of mediation analysis arises when researchers are interested in potential underlying mechanism between an exposure (e.g., DNA methylation) and an outcome (e.g., the survival time and status of cancer patients) and long to acquire an in-depth insight into such understanding. Formally, a mediating variable (or mediator) is defined as an intermediate variable (e.g., gene expression) in the causal sequence that relates the exposure to the outcome [18].

With datasets available from The Cancer Genome Atlas (TCGA) project as an illustrative example [23], in this study we aim to utilize appropriate mediation models to investigate the mechanism of gene expression on the signaling pathway from DNA methylation to the survival risk of cancers. In recent years, there have been a lot of methodology extensions of traditional linear mediation analysis to time-to-event outcome with censored data. Tein and MacKinnon (2003) [24] studied the estimation and inference of mediated effect for survival outcome under the context of the log-survival and log-hazard time models. Lange and Hansen (2011) [25] formulized the natural direct and indirect effects for time-to-event outcome using an additive hazard model within the counterfactual framework and illustrated its application by analyzing socioeconomic status, work environment, and long-term sickness absence. The natural direct and indirect effects were further described using a proportional hazards model with a rare outcome or an accelerated failure time model [26]. Wang and Zhang (2011) [27] proposed a Bayesian Tobit approach to examine the mediation effect for censored data and applied it to study whether verbal memory ability mediated the relationship between age and everyday functioning. More recently, Luo et al. (2020) [28] investigated survival mediation methods with high-dimensional mediators by borrowing the idea of screening-penalization procedure [29]. Although great advances have been achieved (see [30] for a comprehensive review of statistical methods for high-dimensional mediation analysis in the era of high-

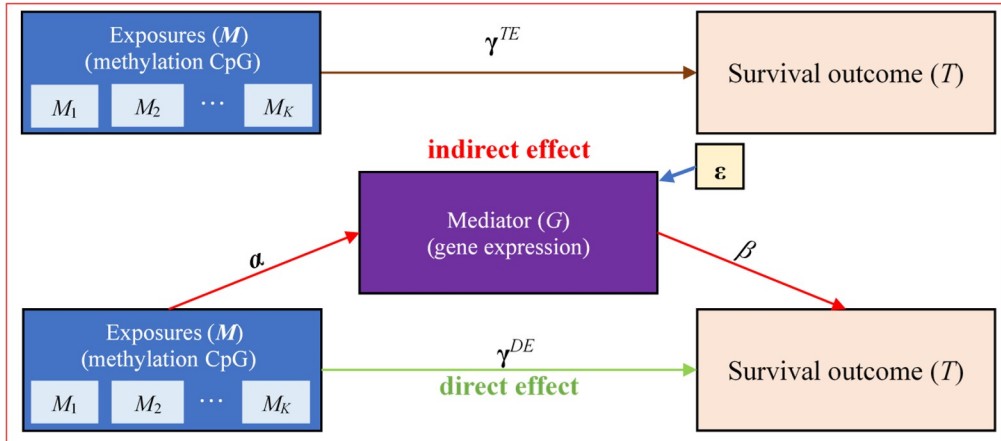

**Fig 1. Statistical framework of the survival mediation analysis with multiple exposures, one mediator and one censored outcome.** The mediation analysis involves in multiple DNA methylations ($M$) as exposures, one gene expression ($G$) as the mediator and one survival time as the outcome ($T$). There are two types of effects from $M$ to $T$: the direct effect of $M$ on $T$ (i.e., $\gamma^{DE}$) and the indirect effect of $M$ on $T$ via the intermediate variable $G$. The indirect effect represents the amount of mediation coming from two sources: the effect from $M$ to $G$ (i.e., $\alpha$) and the effect from $G$ to $T$ (i.e., $\beta$).

throughput genomics), methods for survival mediation analysis are still yet to be developed. These prior methods often consider one or multiple mediators and cannot be readily applied to our setting where multiple exposures, sometimes high-dimensional, would be involved. Specially, we wish to implement a gene-centric mediation analysis to simultaneously model a group of DNA methylation CpG sites as exposures and explore the mediating role of gene expression in the influence of methylations on the survival risk of cancer patients ([Fig 1]).

To effectively examine mediation effect in survival studies with multiple exposures, we first employed the variance component-based score test to assess the association between a group of DNA methylation CpG sites and expression level for each gene under the framework of linear mixed-effects model (lmm) [31–34], and then tested for the effect of gene expression on the overall survival time while adjusting for the direct influence of these methylations within the framework of Cox linear mixed-effects model (coxlmm) [35–38]. This mediation analysis procedure is displayed in [Fig 1], where the methylation-expression effect (i.e., $\alpha$) and the expression-survival effect (i.e., $\beta$) can be causally interpreted if individual mediation models are correctly constructed and identifiability assumptions are satisfied [39–41]. Besides the explicit assumption of temporal ordering between methylation, gene expression and survival outcome, the additional assumptions for causal interpretation of these effects include: (i) the confounding between the methylations and the survival outcome is correctly controlled; (ii) the confounding between the gene expression and the survival outcome is correctly controlled; (iii) the confounding between the methylations and the gene expression is correctly controlled; and (iv) there should be no expression-survival confounders which are themselves affected by the methylations. The above assumptions are also known as sequential ignobility assumptions or no unmeasured confounding assumptions [20,42–45]. Note that, our mediation analysis method can still be employed for identifying potential methylation CpG sites or candidate genes for further exploration, even when the above causality conditions are not completely satisfied.

We subsequently determined the existence of mediation effect to detect candidate genes with potentially mediating roles by adherence to the principle of intersection-union test (IUT) [46–52], in which the maximum of the two $P$-values (denoted by $P_{\max}$) in the two tests above is

taken as the significance measurement. Although it is conceptually straightforward, this naïve IUT-based approach is oftentimes extremely conservative especially in large scale mediation effect tests due to its composite null nature [43,53], which can be equivalently expressed as a combination of three disjoint component null hypotheses. To correct the intrinsic conservativeness of IUT, we estimated the proportion for each component null hypothesis and constructed a novel null distribution for $P_{max}$ by fitting a three-component mixture null distribution [54], which, in contrast to the naïve uniform null distribution of $P_{max}$ assumed in previous literature [51,52,55], can lead to a desirable control of family-wise error rate (FWER) or false discovery rate (FDR). We thus refer to our proposed approach framework as the intersection-union survival mixture-adjusted mediation test (IUSMMT).

Finally, we applied IUSMMT to ten TCGA cancers and identified multiple genes with mediation effects. We revealed that, although DNA methylation CpG sites across the whole genome showed pleiotropic regularization on gene expressions of various cancers, most of the detected genetic regions, in which gene expressions played key roles as active mediators, were cancer type-specific and exhibited full mediations lying in the signaling pathway from DNA methylation CpG sites to the survival risk of cancers. Overall, IUSMMT represents an effective and powerful statistical tool for survival mediation analysis; our results also provide new insights into the functional role of DNA methylation and gene expression in cancer progression/prognosis and demonstrate potential therapeutic targets for future clinical practice.

## Results

### Overview of IUSMMT

We first offer an overview of the proposed IUSMMT method (Fig 2), with more details illustrated in the S1–S4 Texts and the section of Materials and Methods. IUSMMT is a gene-centric mediation method especially for survival data and dedicates to detect candidate genes with potential mediating effects standing on the signaling pathway from DNA methylation CpG sites to cancer survival. It proceeds in the following steps. First, IUSMMT calculates $P$-values of the methylation-expression effect (i.e., $P_{\alpha}$) and the expression-survival effect (i.e., $P_{\beta}$) and takes them as input (Fig 2A); here $P_{\alpha}$ is obtained via a variance component-based score test within lmm by assuming each of $a$ following a mean-zero normal distribution with an unknown variance, and $P_{\beta}$ is yielded through the Wald test within coxlmm. Second, to determine whether a gene of focus has a mediation effect, IUSMMT classifies the joint null hypothesis $H_0$: $\alpha\beta = 0$ into three composite null sub-hypotheses and takes the maximum of $P_{\alpha}$ and $P_{\beta}$ as a significance measurement in terms of the IUT principle (Fig 2B). Finally, to enhance the statistical power, IUSMMT estimates the proportion for each component of the three null hypotheses (Fig 2C) and constructs a newly empirical exact null distribution by fitting a three-component mixture null distribution. Afterwards, an effective control of FWER or FDR is achieved on the basis of the estimated mixture null distribution. As shown below, the power of IUSMMT would improve and the bias in estimated mixture proportions would decrease with the increase of sample sizes.

### Results for simulation studies

### Estimated null proportions

Following the statistical framework of survival mediation analysis shown in Fig 1, we generated expression and survival outcome based on a set of real methylation values of TCGA under various sample sizes and proportion parameters, with details of simulation described in the Materials and Methods section. We first present the results for simulation studies and

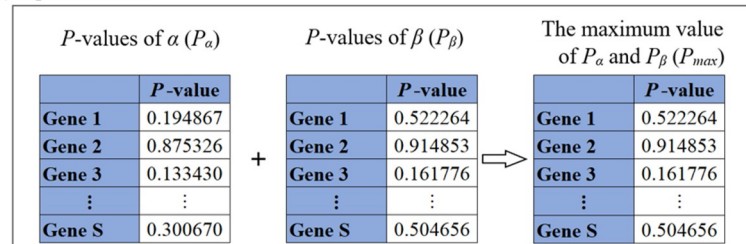

(A) Input matrixes

(B) Three composite null sub-hypotheses

(C) Three-component mixture null distribution

**Fig 2. An overview of IUSMMT for examining the mediation effect in survival model.** Here, $a = (\alpha_1, \ldots, \alpha_K)$ is the vector of effect sizes of a set of DNA methylation CpG sites (i.e., the exposures) on the gene expression level (i.e., the mediator), with $K$ the number of CpG sites within that gene; and $\beta$ is the expression-survival effect; $S$ indicates the total number of genes; $n$ denotes the sample size. (A) IUSMMT first separately evaluates the significance of $a$ and $\beta$, and calculates $P_\alpha$ and $P_\beta$; where $P_\alpha$ is obtained by a variance component-based score test within the linear mixed-effects model by assuming each of $a$ following a mean-zero normal distribution with an unknown variance $\tau_2$, while $P_\beta$ is yielded through the Wald test within the Cox linear mixed-effects model. Then, IUSMMT takes the two $P$-values as input. (B) The hypothesis testing of mediation effect is to examine whether the product of $\alpha$ and $\beta$ is zero or not (i.e., $H_0$: $\alpha\beta = 0$) and can be divided into three composite null sub-hypotheses. (C) In the three-component mixture null distribution, $\kappa_{10}$ stands for the probability that the exposures are related to the mediator in the exposure-mediator model but the mediator is not associated with the survival outcome in the mediator-outcome model; $\kappa_{01}$ stands for the probability that the exposures are not related to the mediator in the exposure-mediator model but the mediator is associated with the survival outcome in the mediator-outcome model; $\kappa_{00}$ stands for the probability that the exposures are not related to the mediator in the exposure-mediator model and the mediator is not associated with the survival outcome in the mediator-outcome model. The definition of other notations used in B and C can be found in the Materials and Methods section.

display estimated proportion parameters for the three-component mixture null distribution under various simulation scenarios in Tables 1 and S1. Totally, the estimates of these proportions show slight to moderate biases in the alternative scenarios but are relatively close to the true values in other cases. Nearly in all scenarios, $\kappa_{00}$ is over-estimated especially when sample size is small. Specifically, in the sparse alternative cases, as can be expected, $\kappa_{10}$ and $\kappa_{01}$ are also over-estimated because they are non-negatively estimated but their true values are actually zero, which necessarily results in the underestimation for $\kappa_{11}$. In contrast, in the dense alternative cases, the opposite patterns are observed, with $\kappa_{00}$ is over-estimated but $\kappa_{01}$ is under-estimated, which is certainly a direct consequence of limited power when testing the effect of the exposure on the mediator (i.e., $\tau_2$) and the effect of the mediator on the survival outcome (i.e., $\beta$) (S1 Fig). Particularly, in all the simulation scenarios, $\kappa_{11}$ is always underestimated or approximately unbiased, implying that we can minimize the false discovery when examining the mediation effect. In addition, we find that the estimates of some proportion parameters (e.g., $\kappa_{00}$) have greater bias under the dense null compared to these under the sparse null. The reason is primarily due to relatively more sufficient information that can be available for estimating these non-zero proportion parameters under the sparse null compared to the dense null (e.g., 90% vs. 10%).

**Table 1. Estimated and true proportion parameters (mean and standard deviation) in the three-component mixture null distribution under the five simulation scenarios with different sample sizes and numbers of mediation tests.**

| dense null | $\kappa_{00} = 0.1$ | $\kappa_{01} = 0.85$ | $\kappa_{10} = 0.05$ | $\kappa_{11} = 0$ |
|---|---|---|---|---|
| n = 250 | 0.441 (0.175) | 0.536 (0.174) | 0.022 (0.012) | 0.001 (0.001) |
| n = 400 | 0.293 (0.155) | 0.675 (0.154) | 0.031 (0.012) | 0.001 (0.001) |
| n = 548 | 0.215 (0.116) | 0.750 (0.116) | 0.034 (0.012) | 0.000 (0.002) |
| sparse null | $\kappa_{00} = 0.99$ | $\kappa_{01} = 0.01$ | $\kappa_{10} = 0$ | $\kappa_{11} = 0$ |
| n = 250 | 0.977 (0.017) | 0.022 (0.017) | 0.000 (0.001) | 0.001 (0.003) |
| n = 400 | 0.986 (0.011) | 0.013 (0.011) | 0.000 (0.001) | 0.001 (0.001) |
| n = 548 | 0.977 (0.018) | 0.023 (0.019) | 0.000 (0.001) | 0.000 (0.000) |
| complete null | $\kappa_{00} = 1$ | $\kappa_{01} = 0$ | $\kappa_{10} = 0$ | $\kappa_{11} = 0$ |
| n = 250 | 0.993 (0.011) | 0.005 (0.010) | 0.002 (0.006) | 0.000 (0.000) |
| n = 400 | 0.998 (0.003) | 0.001 (0.003) | 0.001 (0.002) | 0.000 (0.000) |
| n = 548 | 0.994 (0.013) | 0.004 (0.012) | 0.003 (0.008) | 0.000 (0.000) |
| dense alternative | $\kappa_{00} = 0.1$ | $\kappa_{01} = 0.75$ | $\kappa_{10} = 0.05$ | $\kappa_{11} = 0.1$ |
| n = 250 | 0.394 (0.179) | 0.537 (0.174) | 0.070 (0.036) | 0.000 (0.000) |
| n = 400 | 0.241 (0.145) | 0.667 (0.143) | 0.081 (0.035) | 0.011 (0.022) |
| n = 548 | 0.172 (0.098) | 0.725 (0.097) | 0.077 (0.037) | 0.027 (0.034) |
| sparse alternative | $\kappa_{00} = 0.9$ | $\kappa_{01} = 0$ | $\kappa_{10} = 0$ | $\kappa_{11} = 0.1$ |
| n = 250 | 0.900 (0.021) | 0.055 (0.020) | 0.033 (0.018) | 0.013 (0.018) |
| n = 400 | 0.892 (0.017) | 0.047 (0.026) | 0.025 (0.018) | 0.036 (0.028) |
| n = 548 | 0.870 (0.029) | 0.061 (0.031) | 0.040 (0.026) | 0.028 (0.028) |

Note: $\kappa_{10}$ stands for the probability that the exposures are related to the mediator in the exposure-mediator model but the mediator is not associated with the survival outcome in the mediator-outcome model; $\kappa_{01}$ stands for the probability that the exposures are not related to the mediator in the exposure-mediator model but the mediator is associated with the survival outcome in the mediator-outcome model; $\kappa_{00}$ stands for the probability that the exposures are not related to the mediator in the exposure-mediator model and the mediator is not associated with the survival outcome in the mediator-outcome model; $\kappa_{11}$ stands for the probability of the existence of mediation effects. The number of genes (i.e., the mediators) was set to $10^4$. Note that, here we only present the estimates of one simulation scenario, with more results shown in S1 Table.

Moreover, to evaluate the impact of various numbers of mediators on the estimators of proportions, we implemented an additional simulation with $10^3$ genes (i.e., mediators) with $\kappa_{11} = 0.10$, $\kappa_{10} = 0.75$, $\kappa_{01} = 0.05$, and $\kappa_{00} = 0.10$ (a case that was very close to proportions obtained from our real applications). As can be anticipated, due to more information available (e.g., more genes), it turns out that increasing the number of mediators from $10^3$ to $10^4$ can generally improve the accuracy in estimating these proportion parameters (S2 Table).

**Type I error and power.** Based on these estimated proportion parameters, the mixture null distribution is constructed, and the mediation effect test is implemented. First, when assessing the performance of type I error control, we demonstrate that IUSMMT, which is based on the estimated mixture null distribution, can maintain the type I error correctly across these simulation scenarios (Figs 3 and S2); however, IUT, which utilizes the uniform distribution as its null distribution, is conservative. This conservativeness of IUT indicates it would be underpowered in the detection of significant mediation effect. In the power assessment, due to the adjustment of the mixture null distribution, IUSMMT is much more powerful compared to IUT (Figs 3 and S3). For instance, when n = 400 and $\beta = 0.05$, compared with IUT, IUSMMT has a 0.06, 0.12, 0.07 or 0.11 higher power under the sparse alternative and 0.25, 0.49, 0.45 or 0.30 higher power under the dense alternative when $\tau_2 = 0.02, 0.04, 0.05$ or $0.10$, clearly indicating the benefit of estimating the empirical power function in the mixture null distribution. Moreover, it is clearly shown that IUSMMT has a much more pronounced

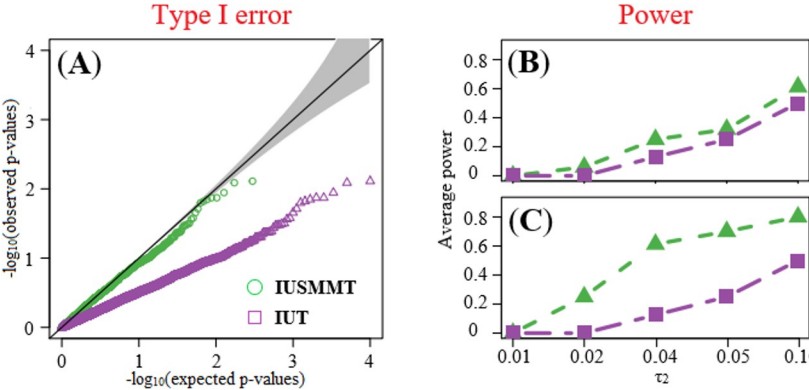

**Fig 3. The QQ plot** (A) **and the average power curve** (B-C) **for IUSMMT and IUT.** The QQ plot (A) is under the scenario of sparse nulls. The average power curve is calculated when the mediation strength parameter $\tau_2$ increases under the sparse alternative (B) and the dense alternative (C). Here, n = 400, $\beta$ = 0.3, and $\tau_2$ = 0.01, 0.02, 0.04, 0.05 or 0.10 at the x-axis. The magnitude of $\tau_2$ quantifies the strength of association between DNA methylation CpG sites and gene expression. In B and C, the number of genes (i.e., mediators) was set to $10^4$.

advantage in power over IUT under the dense alternative (Figs 3 and S3). For example, IUSMMT is on average 29.8% more powerful than IUT under the dense alternative, while is on average only 7.3% more powerful under the sparse alternative.

## DNA methylations are significantly associated with the survival risk of cancers

We now applied IUSMMT to ten types of cancers available from TCGA [23]; summary information of these cancers is shown in S3 and S4 Tables. We first examine the association between a group of DNA methylation CpG sites and the survival risk of cancers for each gene (i.e., $H_0: \gamma^{TE} = 0$). A total of 57 (30 unique) regions of DNA methylation CpG sites are identified to be associated with the overall survival of these cancers (FDR < 0.05) except BLCA and HNSC (Table 2), including 8 methylation regions for BRCA, 7 for CESC, 7 for COAD, 17 for KIRP, 4 for LUAD, 1 for LUSC, 9 for SARC and 4 for STAD. Approximately 23.3 (= 7/30) of these regions of DNA methylation CpG sites exhibit pleiotropic effects (Fig 4A). For example, methylation CpG sites located in *ACAA2*, *NUSAP1*, *OGFOD1*, *PSMD5*, *SNRNP40* and *XRCC6* are simultaneously associated with five cancers including BRCA, CESC, COAD, KIRP and SARC; methylation CpG sites located in *USP37* are simultaneously related to four cancers (i.e., BRCA, CESC, COAD and SARC). Moreover, we recognize some cancer type-specific methylation CpG sites, including all the associated methylation sites identified for LUAD, STAD or LUSC, and partly for KIRP (11 out of 17), SARC (2 out of 9), or BRCA (1 out 8).

## Identified associations between methylation and expression

We here evaluate the association between a group of DNA methylation CpG sites and gene expression (i.e., $H_0: \boldsymbol{\alpha} = 0$) through SKAT with the linear kernel. As a result, a particularly large number of associations are detected (FDR < 0.05) (Table 2), with the number of methylation-regulated genes ranging from 8,182 (62.8% = 8,182/13,029) for KIRP to 11,872 (89.5% = 11,872/13,270) for BRCA and an average of 79.3% across these cancers. There are approximately 98.2% methylation-regulated genes shared by at least two types of cancers, including 3,801 genes (e.g., *HIST1H4F*, *FBP1* and *CPEB4*) shared across all cancers.

**Table 2. Number of associated regions of DNA methylation CpG sites and genes identified in the survival mediation analysis.**

| cancers | $\gamma^{TE}$ | $\alpha$ (%#) | $\beta$ (%#) | $\gamma^{DE}$ (%#) | Mediation effect test | | |
| --- | --- | --- | --- | --- | --- | --- | --- |
| | | | | | IUT (%$) | IUSMMT (%$) | |
| BLCA | 0 | 11,325 (86.29) | 76 (0.58) | 0 (0) | 41 (0.31) | 61 (0.46) | 8 (13.1) |
| BRCA | 8 | 11,872 (89.46) | 56 (0.42) | 8 (0.05) | 30 (0.23) | 46 (0.35) | 14 (30.4) |
| CESC | 7 | 10,059 (76.19) | 9 (0.07) | 8 (0.06) | 1 (0.01) | 1 (0.01) | 0 (0) |
| COAD | 7 | 8,963 (68.25) | 0 (0) | 7 (0.05) | 0 (0) | 0 (0) | 0 (0) |
| HNSC | 0 | 11,284 (84.73) | 42 (0.32) | 0 (0) | 23 (0.17) | 49 (0.37) | 8 (16.7) |
| KIRP | 17 | 8182 (62.80) | 170 (1.30) | 32 (0.22) | 34 (0.26) | 34 (0.26) | 1 (2.9) |
| LUAD | 4 | 10,342 (77.80) | 17 (0.13) | 1 (0.01) | 176 (1.32) | 188 (1.41) | 39 (20.9) |
| LUSC | 1 | 11,427 (84.73) | 3 (0.02) | 0 (0) | 3 (0.02) | 3 (0.02) | 0 (0) |
| SARC | 9 | 10,653 (81.73) | 204 (1.57) | 10 (0.07) | 88 (0.68) | 156 (1.2) | 18 (11.5) |
| STAD | 4 | 10,879 (80.84) | 8 (0.06) | 3 (0.02) | 4 (0.03) | 5 (0.04) | 0 (0) |
| total | 57 | 104,986 | 585 | 69 | 400 | 543 | 88 (16.3) |
| unique | 30 | 13,801 | 570 | 41 | 392 | 529 | |
| pleiotropy (%) | 7 (23.3) | 13,553 (98.2) | 15 (2.6) | 7 (17.1) | 8 (2.0) | 14 (2.6) | |

Note

# denotes the proportion among the total genes under investigation

$ denotes the proportion among the genes that are associated with the survival risk of cancers; the proportion of pleiotropy is computed by the ratio between the number of associations with pleiotropic effects and the number of unique associations. The last second column shows the number and proportion of genes with mediating effects, and the last column shows the number and proportion of potential passenger methylation events among genes with mediating roles.

We also performed the similar methylation-expression analysis in 193 normal tissues combined across the ten cancers and find that 43.9% (= 6075/13840) of genes are methylation-regulated (FDR < 0.05). This proportion is relatively smaller than that obtained from the tumor tissues, which may be a direct consequence of low power due to smaller sample size of normal tissues compared tumor tissues and the distinction mechanism in gene regulation between normal and tumor tissues. On average, 81.0% of methylation-regulated genes discovered in normal tissues are overlapped with these detected in tumor tissues across the cancers (S5A Fig). Interestingly, we observe that the effect of methylation on expression is much stronger in tumor tissues compared with that in the normal tissue. For example, the median value of $\tau_2$, which can be employed to quantify the magnitude of the methylation effect on expression, is 3.7 times higher in the BRCA tissue than that in the normal tissue (S5B Fig).

## Identified associations between expression and survival risk

We next explore the association between the expression level and the survival risk of cancers for each gene (i.e., $H_0: \beta = 0$) while adjusting for the direct effects of methylation alterations within the framework of coxlmm. The number of associated genes varies from zero for COAD to 204 for SARC (FDR < 0.05). Approximately 2.6% of these associated genes are simultaneously shared by at least two cancers (Fig 4B), while most of the associated genes (~97.4%) are cancer type-specific. In addition, a total of 69 (41 unique) regions of methylation CpG sites also exhibit direct influence (i.e., $\gamma^{DE} \neq 0$) on the overall survival of some cancers (except BLCA, HNSC and LUSC) (Table 2 and Fig 4C); and approximately 17.1% of significant regions of methylation CpG sites show direct pleiotropic effects. For example, the methylation regions located in *ACAA2*, *NUSAP1*, *OGFOD1*, *PSMD5*, *SNRNP40*, *USP37* and *XRCC6* are shared by five cancers (i.e., BRCA, CESC, COAD, KIRP and SARC). Moreover, we find that there are 12 methylation regions (i.e., located within *ACAA2*, *NUSAP1*, *OGFOD1*, *PSMD5*,

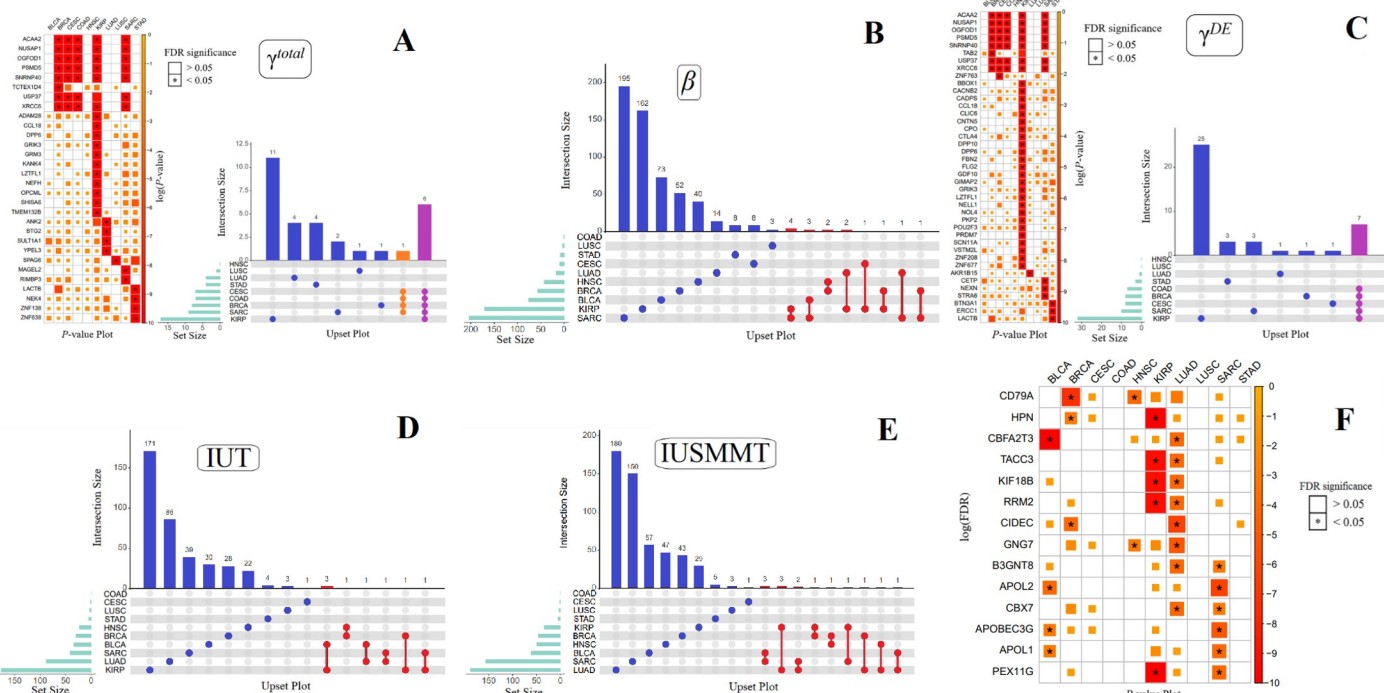

**Fig 4. Upset plot and heatmap plot of the *P*-values (A-F).** In the heatmaps of $\gamma^{TE}$ (A) and $\gamma^{DE}$ (C), the color of each box indicates the magnitude of the *P*-value. The number in the comment part represents the *P*-value processed by the negative logarithmic transformation. The darker the color, the smaller the *P*-value. In the Upset plots of $\gamma^{TE}$, $\beta$, $\gamma^{DE}$, IUT, and IUSMMT, each bar shows the number of shared genes. In these Upset plots, the blue part represents cancer type-specific genes, and the red, green, orange, and purple parts represent genes shared in two, three, four and five types of cancers. (F) The heatmap of the *P*-values of the 14 overlapped genes across all the cancers. The color of each box indicates the magnitude of the *P*-value. The number in the comment part represents the *P*-value processed by the negative log transformation, the darker the color, the smaller the *P*-value.

*SNRNP40*, *USP37*, *XRCC6*, *CCL18*, *DPP6*, *GRIK3*, *LZTFL1*, and *LACTB*) that not only have non-zero total effect on the survival risk, but also have substantial direct influence on the survival risk of cancers.

## Associated genes with mediating effects

We now utilize IUT and IUSMMT to assess whether the expression level of a gene has substantial mediating effect on the pathway from methylation CpG sites to the survival risk of cancers. The estimated proportion parameters for the three-component mixture null distribution for each cancer are shown in Table 3. These proportions also reflect the probability of the association between the methylations and the gene expression (i.e., $\kappa_{10}$), and between the gene expression and the survival time (i.e., $\kappa_{01}$), in line with the results described above. The QQ-plot of $P_{max}$ for each cancer is demonstrated in S4 Fig. An observable upward deviation of $P_{max}$ from the diagonal line implies the presence of mediating genes for most of these cancers except COAD. Specifically, IUSMMT detects 35.8% more genes with mediating effects compared to IUT, and all genes having mediation effects identified by IUT (a total of 400 genes) are also discovered by IUSMMT. Therefore, in the following we focus on the results of IUSMMT. As suggested by the association results described above and in terms of the principle of IUSMMT, except COAD (consistent with the pattern of the distribution of its $P_{max}$ shown in S4 Fig), we identify multiple genes mediating the impact of methylation CpG sites on the survival risk of various cancers (Table 2 and Fig 4E). Specifically, there are a total of 543 (529 unique) significant methylation-expression mediation associations, with the number of genes exhibiting

**Table 3. Estimated proportion parameters in the three-component mixture null distribution for 10 TCGA cancers.**

| cancers | $\kappa_{10}$ | $\kappa_{01}$ | $\kappa_{00}$ | $\kappa_{11}$ |
|---|---|---|---|---|
| BLCA | 0.858 | 0.001 | 0.136 | 0.005 |
| BRCA | 0.891 | 0.001 | 0.105 | 0.004 |
| CESC | 0.761 | 0.000 | 0.238 | 0.001 |
| COAD | 0.683 | 0.000 | 0.317 | 0.000 |
| HNSC | 0.844 | 0.000 | 0.152 | 0.003 |
| KIRP | 0.621 | 0.006 | 0.366 | 0.007 |
| LUAD | 0.777 | 0.000 | 0.222 | 0.001 |
| LUSC | 0.847 | 0.000 | 0.153 | 0.000 |
| SARC | 0.803 | 0.002 | 0.181 | 0.014 |
| STAD | 0.808 | 0.000 | 0.191 | 0.000 |
| average | 0.789 | 0.001 | 0.206 | 0.004 |

Note: $\kappa_{10}$ stands for the probability that the methylations are related to the gene expression in the exposure-mediator model but the gene expression is not associated with the survival outcome in the mediator-outcome model; $\kappa_{01}$ stands for the probability that the methylations are not related to the gene expression in the exposure-mediator model but the gene expression is associated with the survival outcome in the mediator-outcome model; $\kappa_{00}$ stands for the probability that the methylations are not related to the gene expression in the exposure-mediator model and the gene expression is not associated with the survival outcome in the mediator-outcome model.

mediating effects ranging from 1 for CESC, to 156 for SARC and 188 for LUAD. It is very interesting that these mediating genes are more likely cancer type-specific as approximately 97.4% of genes are identified for a single cancer while only 2.6% (i.e., a total of 14 genes, including *CD79A*, *HPN*, *CBFA2T3*, *TACC3*, *KIF18B*, *RRM2*, *CIDEC*, *GNG7*, *B3GNT8*, *APOL2*, *CBX7*, *APOBEC3G*, *APOL1*, and *PEX11G*) are shared across distinct cancers (Fig 4F).

## Functional role of selective DNA methylation and gene expression on TCGA cancers

It first needs to emphasize that among these methylation-expression mediation associations, nearly all are full mediations, with only one being partial mediation (i.e., *ZNF763* for CESC) (S6 Fig). *ZNF763* is a typical zinc finger protein containing Krüppel-associated box (KRAB), which is reportedly related to the development of cervical cancer. In addition, KRAB activates NF-κB by participating in the assembly of the IκB kinase (IKK) complex and promoting phosphorylation of IKK [56], while NF-κB is a multifunctional transcription factor and is related to the occurrence and development of cervical cancer [57]. Taken together, it is suggested that *ZNF763* may contribute to the occurrence of cervical cancer by enhancing NF-κB signaling and changing cell growth. In addition, among the 3,801 methylation-regulated genes that are shared by all analyzed cancers, 39 are histone genes. It has been revealed that histone gene loci are abnormally hypermethylated in a wide range of solid tumors [58,59]. For example, *HIST1H4F*, one of the identified histone genes with mediating effect, was abnormally hypermethylated in 17 types of cancers, acting as a potential universal-cancer-only methylation (UCOM) marker [59].

   Prior studies also offered empirical evidence for these identified mediating genes, three of which are described in detail as follows, with all the mediating genes shown at https://github.com/biostatpzeng/IUSMMT. First, *TRIM26*, as a mediating gene associated with BLCA, was proved to play a key role in several types of cancers. For example, it was demonstrated that

*TRIM26* had an oncogene impact on bladder cancer through regulating cell proliferation, migration and invasion via the Akt/GSK 3 β/β-catenin pathway [60], and methylation CpG sites had a significant effect on the regulation of *TRIM26* [61]. Second, *ZNF496*, as a mediating gene related to BRCA, was shown to significantly suppress ERα transactivation through over-expression, reduce the expression of estrogen receptor-alpha specific target genes, and inhibit the growth of breast cancer cells via ERα in an E2-dependent manner [62]. On the other hand, ERα is a key transcription factor involved in the proliferation and differentiation of mammary epithelia and has been demonstrated as an important predictor of breast cancer prognosis and a therapeutic target [63,64]. Moreover, the ZNF family, to which *ZNF496* belongs, has been shown to be mechanically linked to DNA methylation variability [65]. Third, *TMBIM6*, as a mediating gene discovered for HNSC, played an important role in the progression of laryngeal squamous cell carcinoma as a downstream target of *RBM15*-mediated N6-methyladenosine modification [66]. In addition, N6-methyladenosine methylation modification involved by *RBM15* regulates gene expression of *TMBIM*.

## Enrichment pathway analysis for mediating genes

We performed the gene ontology (GO) and KEGG pathway enrichment analysis for all identified mediating genes using the DAVID database [67] and showed results in S7 Fig. In brief, the GO analysis demonstrates that the biological processes of these genes are concentrated in DNA template, immune response and cell-cell adhesion, the cellular components of these genes mainly involve cytoplasm, cytosol and perinuclear region of cytoplasm, and the molecular functions of these genes are primarily manifested in protein binding, cadherin binding and microtubule binding. The KEGG analysis of these genes also reveals multiple signaling pathways which are related to PI3K-Akt, Ras and histidine metabolism.

## Direction of mediation effects

We further examined the direction of the effect of methylation CpG sites on expression and the effect of expression on the survival risk for these mediating genes (Table 4). It is found that the expression level of most of the genes (68.7% = 373/543) were inhibited by methylation alterations, while the expression level of some genes (~31.3%) was also upregulated by

**Table 4. Direction of the effect of methylations on expression and the effect of expression on the survival risk of cancers for identified genes with mediating influence.**

| cancers | N | Direction ($\alpha$ & $\beta$) | | | |
|---|---|---|---|---|---|
| | | + + | - - | - + | + - |
| BLCA | 61 | 12 | 24 | 12 | 13 |
| BRCA | 46 | 2 | 28 | 4 | 12 |
| CESC | 1 | 1 | 0 | 0 | 0 |
| HNSC | 49 | 2 | 18 | 3 | 26 |
| KIRP | 34 | 5 | 11 | 16 | 2 |
| LUAD | 188 | 19 | 70 | 59 | 40 |
| LUSC | 3 | 2 | 0 | 1 | 0 |
| SARC | 156 | 8 | 59 | 64 | 25 |
| STAD | 5 | 1 | 1 | 3 | 0 |
| total | 543 | 52 | 211 | 162 | 118 |

Note: *N* denotes the number of genes with mediating effect; + or—refers to the positive or negative direction in the effect of methylations on expression (i.e., $\alpha$) or in the effect of expression on the survival risk of cancers ($\beta$).

methylation alterations, in line with prior observations of the dual function of methylations on gene expression [68, 69]. The downregulated expressions may lead to higher (43.4% = 162/373) or lower (56.6% = 211/373) survival risk of cancers, and the upregulated expressions can also likely result in higher (30.6% = 52/170) or lower (68.8% = 118/170) survival risk of cancers. Totally, 48.4% of (= 263/543) genes have the methylation-expression effect and the expression-survival effect in the same direction and 51.6% (= 280/543) of genes have opposite directions.

### Distinguish passenger methylation events from mediation methylation events

Finally, we highlight that the effect pathway in our mediation analysis depends on a critical assumption that DNA methylation CpG sites regulate gene expression rather than conversely. However, such assumption may not fully hold as recent methylation studies in large cancer datasets have revealed that a bulk of methylation alterations observed in cancers might be a consequence of global epigenetic remodeling mechanism including (i) global loss via replication related errors; and (ii) CpG island/shore methylation gain associated with tumor proliferation [70,71]. Under this case, altered expression of a handful of key genes would have a reverse influence on DNA methylation CpG sites over the cancer epigenome. Namely, the expression of a gene can affect the survival risk of cancers while impacting DNA methylations, leading to the biological consequence that changed DNA methylations are just passenger events.

Therefore, distinguishing passenger methylation events from mediation methylation events that are likely to have a direct or indirect (via expression mediation) influence on the survival risk of cancers is necessary for the biological interpretation of results in our mediation analysis. In terms of the result of multivariate variance-component score test conducted via MultiSKAT [72], we find that approximately 16.3% (= 88/543) of the discovered mediation methylation events might be passenger methylation events across these cancers after the adjustment with Bonferroni's method ($P < 0.05$) (Table 2).

## Discussions

In the present study we have proposed a novel statistical approach, called IUSMMT, for examining high-dimensional mediation effects in survival models with multiple exposures and one mediator. We have applied IUSMMT to ten TCGA cancers to identify genes that likely exhibited mediation effects of gene expression on the signaling pathway from DNA methylation CpG sites to the survival risk of cancers and observed some interesting results which provide supplementary insights on the biological mechanism from DNA methylation to gene expression and to survival outcome. First, only a few of DNA methylation CpG sites showed a total impact on the survival risk of cancers, although we cannot completely rule out the possibility of low statistical power due to small sample sizes and high censoring rates of these analyzed cancers. Second, we found that DNA methylation CpG sites can influence expression level directly for a wide range of genes and that most of these methylation-regulated genes were shared across distinct cancers, authenticating the prior finding that DNA methylation is a pervasive epigenomic mechanism of gene regulation [73,74].

Third, many genes were related to the survival risk of cancers, whereas most of them (~84.6%) were cancer type-specific, with only a few simultaneously shared between various cancers. As a result, although a wealth of genes mediated the influence of DNA methylation CpG sites on the overall survival time of cancers, the majority of the mediations associations were also cancer type-specific. This specificity largely reflects the inter-tumor heterogeneity of

the transcriptomic influence on cancers and the tissue distinction in cancer prognosis [75], implying that mediated pathways of the epigenomic impact vary cancer by cancer although DNA methylation pervasively regulates gene expression. It also has important implication on targeted therapies of precision cancers medicine by intervening DNA methylations [76–78].

Fourth, besides the indirect impact mediated by genes, multiple regions of DNA methylation CpG sites also have direct effects on these cancers. Fifth, we found that almost all the associations were full mediations, with only few partial mediations. Besides the truly biological mechanism, other possible explanations also include low power in examining the direct effects (i.e., $\gamma^{DE}$) of DNA methylation CpG sites for these TCGA cancers. Note that, potentially due to the lack of coverage of distal regulatory elements in the TCGA 450K chip, our analysis cannot include distal enhancer when defining methylation loci for every gene although we can consider methylation alterations around the promoter. It has been shown that the distal enhancers of CpG sites are unmethylated in normal cells but often gain methylations in cancer cells and tissues [79,80]. With the growing increase in the use of whole genome bisulfite whole genome sequencing (WGBS) and Illumina's newly released methylated EPIC BeadChip [81], including methylation loci around the enhancer would become feasible and important in epigenetic mediation analysis from both the biological and statistical perspectives. For instance, a mediation event, which was identified as partial mediation using only promoter CpG sites, may be tagged as fully mediated if enhancers/transcription factor binding would be considered as well. This is certainly an interesting topic for future work. It also needs to emphasize that our analysis cannot directly indicate these mediating genes include in proliferation or cell cycle related genes. Further experimental studies are warranted to elucidate the biological function of these genes.

In addition, although many studies have demonstrated that the variability in DNA methylation level can be largely attributable to diverse cell types and that cellular composition can be an important factor for elucidating biological processes [82] (e.g., the ratio of infiltrating neutrophils to plasma cells has important prognostic significance in cancer survival [83]), in the present analysis we followed prior work (e.g., [43,84]) and did not consider cell types as covariates due to two main reasons listed below. First, because cell types are often inferred directly from gene expressions or methylations [85–87], we did not incorporate them in the mediation model to avoid using data twice. Second, to our knowledge, adjusting cell types is primarily for DNA in the blood sample, and is very rare in the tumor tissue. Nevertheless, we recognize the critical role of cell types in the high-dimensional mediation analysis [88] and would pursue this issue in our future work.

It needs to highlight that IUSMMT adopted the principle of intersection-union test and is conceptually straightforward [47,49,50]. Like many previous mediation analyses, we implemented IUSMMT in two stages. Specifically, in the first step, the influence of DNA methylation CpG sites on the gene expression was examined and in the second step the association between the expression levels was investigated. Note that, although there are two null hypotheses involved in IUSMMT, no adjustment for multiplicity is needed because the overall null hypothesis of no mediation effect is rejected if and only if each of individual null hypotheses (rather than any of individual null hypotheses) is rejected. It is evident that the power of IUSMMT depends on the individual power in each stage.

Our analysis relies on an implicit assumption that DNA methylation CpG sites can regulate gene expression [89–92]. This assumption is largely supported by prior finding that epigenetic modification is closely relevant to the gene expression level which in turn has a direct function consequence on complex human diseases including malignant tumors [93,94]. Previous studies also showed that genes often exhibited cancer type-specific methylation changes and contributed to a higher incidence in cancer patients [95]. In addition, methylation CpG sites not

only affect the survival through the expression of its located genes but also work via a variety of other mechanisms, including splice variants and enhancer regions [96–98]. Recent work discovered that the methylation profiles of patients of four cancer subtypes played an important role in regulating gene expression during many biological processes and identified some functional genes with different methylation status in different subtypes [99].

However, the regulation assumption from DNA methylations to gene expression may not fully hold as revealed in terms of recent studies which found that methylation alterations can be reversely regulated by gene expression under the global epigenetic remodeling mechanism [70,71]. As a consequence, altered DNA methylation may simply represent a passenger event rather than a mediation event. To distinguish passenger methylation events from mediation methylation events, using the reverse regression analysis we found statistically supportive evidence that a small fraction of mediation associations may be passenger methylation events. However, this reverse analysis strategy is an *ad-hoc* approach which has no clear theoretical basis and which only considered the *cis* influence of gene expression regulation on methylation. It also demonstrates an important limitation of mediation analysis; namely, although it is useful for uncovering evidence of causal association, mediation analysis *per se* can be inadequate for completely determining the direction of the identified causal relationship. Therefore, addressing passenger methylation event in our epigenetic mediation analysis comprehensively is a promising but challenging direction for future investigation.

We finally highlight that there is still some room for further enhancement of the power in the first stage where the relationship between a set of DNA methylation CpG sites and gene expression level was examined with the variance-component test with a linear kernel function in the present study. As well-documented in previous studies, other more complicated kernels or a composite optimal kernel may have the potential to improve the power [100–106]. In addition, herein we implicitly suppose that all DNA methylation CpG sites had an influence on the gene expression, which may not hold in the real-life dataset. Instead, there may be only a few DNA methylation CpG sites regulating the expression level of a gene. Under this circumstance, a sparse relationship between DNA methylation CpG sites and the expression level should be modeled [51,107]. Furthermore, IUSMMT may be sub-optimal if the effect sizes of DNA methylation CpG sites located within the gene have the same direction (e.g., all are positive or negative). In this scenario, the burden test with a weighted score across these methylation CpG sites often leads to more powerful mediation approaches [33,101,108]. However, it seems very challenging to select a test that is consistently optimal across various settings because of the true relationship is unknown. Alternatively, it is hence desirable to construct a feasible omnibus mediation effect test that can aggregate all these strengths stated above. Nevertheless, in the present study we offer a very general framework for the assessment of mediation effect and we leave these issues mentioned above as an important and promising research avenue in our further work.

## Materials and methods

### Modeling framework for survival mediation analysis with multiple exposures

IUSMMT is a gene-centric high-dimensional survival mediation approach and considers one gene at each time. Assume there are multiple exposures ($M$; e.g., a set of DNA methylation values for CpG sites located within a given gene), one mediator ($G$; e.g., expression level of that gene) and one survival outcome ($T$; including the survival time $t$ and the survival status $d$) for $n$ individuals. In the conventional mediation analysis [18–22], the

impact of $M$ on $T$ stands for the direct effect, and the influence of $M$ on $G$ and subsequently $G$ on $T$ for the indirect effect. Our objective is to evaluate the causal effect of $M$ on $T$ that is mediated via $G$. If $M$ affects $T$ only through $G$ (i.e., $\gamma^{DE} = 0$), it is called full mediation; otherwise, it is referred to as partial mediation if $\gamma^{DE} \neq 0$ [109]. For a gene under investigation, the associations between DNA methylations, expression, or clinical covariates ($X$) and the survival outcome can be determined within the framework of mediation analysis through the following procedures.

## Step 1: Cox linear mixed-effects model testing for the total effect of methylations on the survival outcome

Following previous work [28,38], we first fit an exposure-outcome Cox model (i.e., methylation-survival model) to examine the association between a group of DNA methylation CpG sites ($M$) of a gene of focus and the survival outcome ($T$) (i.e., $M \rightarrow T$) while adjusting for the impact of existing covariates ($X$)

$$\log(h(t|\boldsymbol{M}, \boldsymbol{X})/h_0(t)) = \sum_{k=1}^{K} M_k \gamma_k^{TE} + \boldsymbol{X}\boldsymbol{w}_1 = \boldsymbol{M}\boldsymbol{\gamma}^{TE} + \boldsymbol{X}\boldsymbol{w}_1 \tag{1}$$

where $h(t|M,X)$ is the hazard risk at time $t$ given $M$ and $X$; $\boldsymbol{\gamma}^{TE} = \left(\gamma_1^{TE}, \ldots, \gamma_K^{TE}\right)$ is the vector of total effect sizes of exposures on the survival outcome; $\boldsymbol{w}_1 = (w_{11}, \ldots, w_{1L})$ denotes the vector of effect sizes of $L$ covariates in the exposure-outcome model; $h_0(t)$ is an arbitrary baseline hazard function, and $K$ is the number of methylations. Nota bene, $K$ varies gene by gene. In terms of the weighted residual method proposed in [110], we find that the proportional hazards assumption required by the utilization of the Cox model (here only covariates are considered; see below) is satisfied across all the ten TCGA cancers ($P>0.05$) (S5 Table).

To assess the relationship between $M$ and $T$ ($H_0$: $\boldsymbol{\gamma}^{TE} = 0$), one may treat $\boldsymbol{\gamma}^{TE}$ to be fixed effects and apply the classical score test (or multivariate Wald test) for hypothesis testing. However, as the number of methylations (i.e., $K$) in a gene might be very large up to several hundred and highly correlated with each other (S8 Fig), the fixed-effects test methods would have a large degree of freedom and are thus underpowered [31–33,111]. Alternatively, we assume $\gamma_k^{TE}(k = 1, \ldots, K)$ follows a normal distribution $N(0, \tau_1)$ in terms of previous relevant studies [32–34], leading to the so-called coxlmm [35,38]. Based on this effect assumption, we estimate and test the direct path within the framework of kernel-machine (KM) based Cox model relying on another equivalent null hypothesis $H_0$: $\tau_1 = 0$ via the coxKM package [112].

Traditionally, the presence of a significant total effect (i.e., $\boldsymbol{\gamma}^{TE} \neq 0$) is the prerequisite for the subsequent mediation test [22]. However, in practice it is not uncommon for the situation where the total effect is nonsignificant (i.e., $\boldsymbol{\gamma}^{TE} = 0$) but a substantial mediation effect remains [113]. Therefore, we always further investigate the existence of mediation effect irrespective of the significance or insignificance of $\boldsymbol{\gamma}^{TE}$.

## Step 2: Linear mixed-effects model testing for the effect of methylations on gene expression

Next, in the exposure-mediator model (i.e., methylation-expression model) we evaluate the association path from methylation CpG sites to expression while controlling the influences of other covariates (i.e., $M \rightarrow G$). With the similar reason described in the first step, like $\gamma^{TE}$ we

also suppose the effects of methylations $\boldsymbol{\alpha}$ have a normal distribution with variance $\tau_2$

$$G = \sum_{k=1}^{K} M_k \alpha_k + \boldsymbol{X} \boldsymbol{w}_2 + \boldsymbol{\varepsilon} = \boldsymbol{M} \boldsymbol{\alpha} + \boldsymbol{X} \boldsymbol{w}_2 + \boldsymbol{\varepsilon}$$

$$\alpha_k \sim N(0, \tau_2)$$

$$\boldsymbol{\varepsilon} \sim N(0, \sigma_\varepsilon^2)$$

(2)

where $\boldsymbol{\alpha} = (\alpha_1, \ldots, \alpha_K)$ is the vector of effect sizes of exposures on the mediator; $\boldsymbol{w}_2 = (w_{21}, \ldots, w_{2L})$ denotes the vector of the effect sizes of covariates in the exposure-mediator model; and $\boldsymbol{\varepsilon}$ is a mean-zero normal residual with variance $\sigma_\varepsilon^2$. We examine the null hypothesis $H_0$: $\boldsymbol{\alpha} = 0$ (or equivalently $\tau_2 = 0$) by utilizing the variance component-based score test within the framework of lmm [31,32,114–116] albeit the likelihood ratio-based test is also applicable [33,117,118]

$$Q = \sum_{k=1}^{K} \left\{ \sum_{i=1}^{n} \left[ M_{ik} (G_i - \boldsymbol{X}_i \hat{\boldsymbol{w}}_2) \right]^2 \right\}$$

(3)

where $\hat{\boldsymbol{w}}_2$ is the estimate of $\boldsymbol{w}_2$ under the null model (i.e., $G = \boldsymbol{X} \boldsymbol{w}_2 + \boldsymbol{\varepsilon}$). Under $H_0$, the score statistic $Q$ follows a mixture of chi-square distribution and the $P$-value is approximately obtained by Davies' method [32,119]. We implement this test via the SKAT package [32].

## Step 3: Cox linear mixed-effects model testing for the effect of gene expression on the survival outcome

In the third step under the mediator-outcome model (i.e., expression-survival model), we examine the path from gene expression ($G$) to the survival outcome ($T$) conditional on methylation CpG sites (i.e., $G \rightarrow T$) through coxlmm

$$\log(h(t|\boldsymbol{M}, G, \boldsymbol{X})/h_0(t)) = \sum_{k=1}^{K} \boldsymbol{M}_k \gamma_k^{DE} + G\beta + \boldsymbol{X} \boldsymbol{w}_3 = \boldsymbol{M} \boldsymbol{\gamma}^{DE} + G\beta + \boldsymbol{X} \boldsymbol{w}_3$$

(4)

where $\boldsymbol{\gamma}^{TE} = (\gamma_1^{TE}, \ldots, \gamma_K^{TE})$ is the vector of the direct effects of methylations, $\beta$ is the effect size of gene expression, and $\boldsymbol{w}_3 = (w_{31}, \ldots, w_{3L})$ denotes the vector of the effect sizes of covariates in the mediator-outcome model. In the same principle, we suppose $\gamma_k^{DE}(k = 1, \ldots, K)$ follows a normal distribution $N(0, \tau_3)$. Herein, we are interested in testing $H_0$: $\beta = 0$. We estimate $\beta$ and implement a Wald test via the coxme package [35,120].

Based on these models described above, the average nature direct effect and nature indirect effect can then be derived (S1 Text) [26,121]; the modeling assumptions required for estimation identifiability and causal interpretation of these effects are also given (S1 Text).

## Intersection-union survival mixture-adjusted mediation test (IUSMMT)

Finally, to verify whether a given gene has mediation effect on the path from methylation CpG sites to the survival risk, we test the joint null hypothesis in which both effects have to be zero: $H_0$: $\boldsymbol{\alpha} = \beta = 0$ (or equivalently, $H_0$: $\tau_2 = \beta = 0$). As mentioned before, we need to implement the similar hypothesis testing across the whole genome. Therefore, this is a high-dimensional problem of mediation effect test. That is, we have $H_{0j}$ ($j = 1, \ldots, S$) for all $S$ genes. For simplicity, in the following we here ignore the subscript $j$. The individual mediation effect test is also equivalent to the hypothesis testing whether the mediation effect exists or not ($H_0$: $\boldsymbol{\alpha}\beta = \boldsymbol{0}$ versus $H_1$: $\boldsymbol{\alpha}\beta \neq \boldsymbol{0}$) in the absence of interactions between the exposures and the mediator and

can be divided into three composite null sub-hypotheses

$$H_0 = \begin{cases} H_{10} : \boldsymbol{\alpha} \neq 0 (\tau_2 \neq 0) \text{ and } \beta = 0 \\ H_{01} : \boldsymbol{\alpha} = 0 (\tau_2 = 0) \text{ and } \beta \neq 0 \\ H_{00} : \boldsymbol{\alpha} = 0 (\tau_2 = 0) \text{ and } \beta = 0 \end{cases} \tag{5}$$

The hypothesis in (5) can be formulated within the framework of the intersection-union test (IUT) (S2 Text) [46–50]

$$H_0 = H_{10} \cup H_{01} \cup H_{00} \text{ versus } H_1 = H_{10}^c \cap H_{01}^c \cap H_{00}^c \tag{6}$$

with $A$ denoting the complement of set $A$.

To conduct IUT, we additionally define $H_0^{\boldsymbol{\alpha}} : \boldsymbol{\alpha} = 0$ versus $H_1^{\boldsymbol{\alpha}} : \boldsymbol{\alpha} \neq 0$ and $H_0^{\boldsymbol{\beta}} : \boldsymbol{\beta} = 0$ versus $H_1^{\boldsymbol{\beta}} : \boldsymbol{\beta} \neq 0$, and suppose the $P$-value obtained from $H_0^{\boldsymbol{\alpha}}$ is $P_{\boldsymbol{\alpha}}$ and the $P$-value obtained from $H_0^{\beta}$ is $P_{\beta}$. To ensure the desirable type I error control of IUT, we exploit $P_{\max} = \max(P_{\boldsymbol{\alpha}}, P_{\beta})$ to evaluate the overall significance of the mediation effect. IUT is advantageous in that the resulting mediation effect test has conceptual simplicity and feasibility of maneuver, and $P_{\max}$ *per se* can be employed as the $P$-value for assessing the significance of mediation [51,52,55,122]. Obviously, when rejecting $H_0$ if $P_{\max}$ is less than the given significance level of $\alpha$, IUT is indeed a level-$\alpha$ test, representing that the type I error of IUT is guaranteed at most $\alpha$ once the rejection decision for $H_0$ is made [47,49].

However, IUT is oftentimes extremely conservative especially when both $H_0^{\boldsymbol{\alpha}}$ and $H_0^{\beta}$ (i.e., $H_{00}$) hold [43,53], which is particularly true in genome-wide mediation studies where most of molecular markers such as gene expression or DNA methylation may be not expected to be related to the outcome of interest. To our knowledge, the commonly-used Sobel test [123,124], which is also often overly conservative [53,125], cannot be directly applicable for examining gene-centric high-dimensional mediation effects. Alternatively, to efficiently correct this conservativeness of IUT, we estimate the proportion for each component of the three null hypotheses and construct a new empirical null distribution for $P_{\max}$ by fitting a three-component mixture null distribution (S3 Text) [54]

$$\begin{aligned} \Pr(P_{\max} \leq u | H_0) &= \kappa_{10} p_{10} u + \kappa_{01} p_{01} u + \kappa_{00} u^2 \\ p_{10} &= \Pr(P_a \leq u | H_{10}) \\ p_{01} &= \Pr(P_{\beta} \leq u | H_{01}) \end{aligned} \tag{7}$$

where $\kappa_{10}$, $\kappa_{01}$ and $\kappa_{00}$ are the proportions corresponding to the three null hypotheses illustrated in (5), $u$ is a given threshold value for the significance evaluation, while $p_{10}$ and $p_{01}$ are actually the power of rejecting $\boldsymbol{a} = 0$ under $H_{10}$ or $\beta = 0$ under $H_{01}$ respectively; and can be estimated via the Grenander method [126]. The estimation of these proportion parameters required in (7) can be easily implemented with methods that were well-established in the FDR literature (S4 Text) [29,127–135]. Once the estimates of these proportions are obtained, the estimated mixture null distribution for $P_{\max}$ can be built to control FWER or FDR. A comprehensive theoretical derivation with regards to the control of FWER or FDR can be conferred in [54].

To distinguish from the naïve IUT-based mediation test which takes $P_{\max}$ as the statistic and the uniform as the null distribution, we refer to the proposed approach as IUSMMT (intersection-union survival mixture-adjusted mediation test) which exploits the estimated mixture as the null distribution. IUSMMT is freely available at https://github.com/biostatpzeng/IUSMMT.

## Simulation studies for type I error control and power evaluation

To evaluate the performance of IUSMMT, we undertake extensive simulations to investigate the type I error control and power. To simulate the truth in practice, we generated the gene expression level ($G$) and the survival outcome with 50 methylation CpG sites ($M$) of *B3GALT4* on 548 BRCA individuals in TCGA [23]. Of note, the selection of this gene and this cancer was primarily because the number of methylation CpG sites and the sample size satisfied our simulation settings. Specifically, we first simulated $G$ via a linear mixed-effect model with the number of methylation CpG sites varying in terms of a uniform distribution (ranging from 10 to 30, with an average of 20). Two covariates $x_1$ (based on the age of the BRCA patients) and $x_2$ (based on the cancer stage of the BRCA patients) ($X$) were also included, both having an effect size of 0.5 in the exposure-mediator model

$$G = \sum_{k=1}^{K} M_k \alpha_k + 0.5 x_1 + 0.5 x_2 + \varepsilon = M\alpha + 0.5 x_1 + 0.5 x_2 + \varepsilon$$

$$\alpha_k \sim N(0, \tau_2)$$

$$\varepsilon_1 \sim N(0, 1)$$

$$(8)$$

Thereafter, we employed the inverse probability method to create the survival time (i.e. $t$) in terms of the Weibull distribution with a fixed shape parameter $\lambda = 1$ and a fixed scale parameter $\rho = 0.01$ [136]. The location parameter of the Weibull distribution was determined by $M$, $G$ and $X$ in the mediator-outcome model

$$\log t = \frac{1}{\rho} \log(-\log(u)/(\lambda \exp(\eta)))$$

$$\eta = \sum_{k=1}^{K} M_k \gamma_k^{DE} + G\beta + 0.5 x_1 + 0.5 x_2 = M\gamma^{DE} + G\beta + 0.5 x_1 + 0.5 x_2$$

$$u \sim U(0, 1)$$

$$\gamma^{DE} \sim N(0, \tau_3)$$

$$(9)$$

where $u$ was a 0–1 uniform variable and $\tau_3 = 0.02$. The censored rate was fixed to be 50% in a random manner (the high censored rate corresponded to the similar situation observed in TCGA cancer dataset; see below).

To balance the statistical power, we set $\tau_2 = 0, 0.01, 0.02, 0.04, 0.05$ or $0.10$ and $\beta = 0, 0.15, 0.2, 0.25$ or $0.30$, and considered various configurations of the two parameters. For each configuration, we set the sample size n = 250, 400 or 548, and generated $10^4$ datasets with $M$, G, $X$, and $T$; that is, we had $S = 10^4$ genes (i.e., mediators). Because there were too many possibilities for the effect sizes and the mixture proportions, in the present study we were primarily interested in several cases that matched closely to our context application. Following the previous study [54], five scenarios for proportion parameters were designed (S1 Table), including the dense null under which $\kappa_{11}$ was set to zero but $\kappa_{10}$ or (and) $\kappa_{10}$ had a small value, the sparse null under which $\kappa_{11}$ was set to zero, but in contrast to the sparse nulls, $\kappa_{00}$ had a small value, and both $\kappa_{10}$ and $\kappa_{10}$ had a relatively large value, the complete null under which $\kappa_{00}$ was set to one, the sparse alternative under which $\kappa_{11}$ was not equal to zero, $\kappa_{00}$ had a large value, and $\kappa_{10}$ and $\kappa_{10}$ were set to zero, and the dense alternative under which $\kappa_{11}$ was not equal to zero, but in contrast to the sparse alternative. The control of type I error was assessed with QQ plot and the power was calculated by the proportion of true positive discoveries among the true mediation effects. We repeated the simulations 100 times and took the average across the replications.

## Application to ten TCGA cancer datasets

We finally applied IUSMMT to ten cancer datasets (S3 Table) publicly available from TCGA [23]. For these cancers, we downloaded their clinical information, RNAseq expressions as well as DNA methylations from UCSC Xena (https://xenabrowser.net/). DNA methylation levels were determined by Illumina Infinium HumanMethylation 450K platform and gene expression profiles were measured via the Illumina HiSeq 2000 RNA Sequencing platform. For methylation measurements, we removed non-CpG sites (i.e., these probes with *ch* labels) and CpG sites located on sex chromosomes; we also excluded cross-reactive probes as suggested in [137]. The beta values of methylation levels were logit-transformed to obtain the M-value for each CpG locus as the M-value is not bounded between zero and one and is thus more valid for our subsequent statistical analysis [138]. For each cancer, we focused on patients of self-report European ancestry while excluding patients whose tissues were formalin-fixed and paraffin-embedded. For gene expression measurements, we only considered protein coding genes and removed genes with over 50% zero expressions and variances smaller than 20% quantile of expression values. Then, we yielded methylation-expression pair for each gene in terms of the position annotation file provided by UCSC Xena. In each gene pair, we standardized both DNA methylation and expression level so that methylation and expression have a mean zero and variance one. According to TCGA gene annotation mapping file, we defined in our analysis methylations as those located within the entire gene body and a 500 bps upstream of the transcription start site (TSS) so that the promoter can be included. The distance of 500 bps is to some extent arbitrary and experience-based. In fact, the optimal extension distance upstream of TSS is not clear and various choices were applied in prior literature [139]. To examine the robustness of various extension distances, we performed a sensitivity analysis by incorporating methylation CpG sites within distinct extended regions, and found that the extension of the distance upstream of TSS in defining methylation loci seems to be very robust and has little influence on our final identification of mediating genes for methylation levels measured with the 450K platform (S9 Fig).

In addition, some clinical covariates available with only a few missing values, such as gender (coded as 0 or 1), age (treated as continuous variable), stage (coded from 1 to 5 and treated as continuous variable), estrogen receptor status (ER) (coded as 0 or 1) and progesterone receptor status (PR) (coded as 0 or 1), were also considered. We had to ignore other common covariates (e.g., alcohol consumption) that had too many missing values although they may be also greatly important. For the remaining datasets, the missing values were simply imputed by the mean if any. Notably, some of these covariates were cancer type-specific (e.g., ER and PR). Following previous studies [38,140–143], we employed the overall survival time and the corresponding survival status as the outcome as there was minimal ambiguity in defining an overall survival event [144]. In brief, overall survival in TCGA was the duration from the diagnosis of cancer to the death of patients. The employed TCGA cancer datasets and data process are summarized in S2 and S4 Tables.

## Identify passenger methylation event via MultiSKAT

To distinguish passenger methylation events from mediation methylation events that are likely to have a direct or indirect (via expression mediation) influence on the survival risk of cancers, we here assume the gene expression may affect a set of DNA methylations and perform a reverse analysis by regressing methylation measurements on expression for all the identified mediating genes based on a multivariate model

$$\boldsymbol{M} = \sum_{k=1}^{K} G\tilde{\beta}_k + \boldsymbol{X}\tilde{\mathbf{W}}_2 + \tilde{\boldsymbol{\varepsilon}} = G\tilde{\boldsymbol{\beta}} + \boldsymbol{X}\tilde{\mathbf{W}}_2 + \tilde{\boldsymbol{\varepsilon}} \tag{10}$$

where $\tilde{\boldsymbol{\beta}} = (\tilde{\beta}_1, \ldots, \tilde{\beta}_K)$ denotes the vector of effect sizes of gene expression on DNA methylation sites, $\tilde{\mathbf{W}}_2$ is the matrix of effect sizes for covariates, and $\tilde{\boldsymbol{\varepsilon}}$ is the matrix of residual error terms. We further suppose that each $\tilde{\beta}_k$ $(k = 1, \ldots, K)$ follows a normal distribution with mean zero and variance $\tau$. Then, our objective is to test for the null hypothesis $H_0$: $\tilde{\boldsymbol{\beta}} = 0$, which is equivalent to examining $H_0$: $\tau = 0$. The corresponding variance component score test statistic is given as

$$Q_M = \{G^T(\boldsymbol{M} - \hat{\boldsymbol{\mu}})\hat{\mathbf{V}}^{-1}\}^T\{G^T(\boldsymbol{M} - \hat{\boldsymbol{\mu}})\hat{\mathbf{V}}^{-1}\} \tag{11}$$

where $\hat{\boldsymbol{\mu}}$ and $\hat{\mathbf{V}}$ are the estimated mean and covariance of $\boldsymbol{M}$ under the null hypothesis. Then, the test statistic $Q_M$ follows a mixture of chi-square distribution and this test can be implemented via MultiSKAT [72].

## Supporting information

**S1 Fig. Estimated power for $\alpha$ and $\beta$.** These powers are estimated under six alternative simulation scenarios from A to I with various mixture proportions. The graph in the left column is the power of $\beta$, and the x-axis is the value of $\beta = 0.15, 0.20, 0.25$ or $0.30$; The graph in the left column is the power of $\alpha$, and the x-axis is the value of $\tau_2 = 0.01, 0.02, 0.04, 0.05$ or $0.10$. These powers are estimated by the average across the 100 replications.
(TIF)

**S2 Fig. QQ plot for IUSMMT and IUT under various the scenarios of nulls and sample sizes.** Here (A), (B) and (C) represent the sample size n = 250, 400 and 548, respectively.
(TIF)

**S3 Fig. Estimated power for IUSMMT and the IUT method.** Here, $\tau_2 = 0.01, 0.02, 0.04, 0.05$ or $0.10$ at the x-axis, β = 0.15, 0.20, 0.25 or 0.30 on the top. These powers are estimated for the nine alternative simulation scenarios from A to I with various values for the mixture proportions by the average across the 100 replications. (A), (B) and (C) represent the sample size n = 250, 400 and 548, respectively.
(TIF)

**S4 Fig. QQ plot for the mediation effect test statistic $P_{\max}$ for 10 TCGA cancers.**
(TIF)

**S5 Fig.** (A) **Proportion of overlapped methylation-regulated genes discovered in various cancer tumor tissues and these discovered in normal tissue.** (B) **Estimated values of $\tau_2$ in the BRCA tissue and in the normal tissue.** Here, $\tau_2$ can be employed to quantify the magnitude of the methylation effect on expression, the sample size for the normal tissue is combined and analyzed across all the 10 cancers.
(TIF)

**S6 Fig. Partial mediation framework of *ZNF763* for CESC.** Please refer to Fig 1 for the interpretation of these parameters shown herein.
(TIF)

**S7 Fig. Results of KEGG pathway enrichment analysis and the significant terms identified by GO enrichment analysis for the genes.**
(TIF)

**S8 Fig. Number of methylation markers belonging to each gene across the while genome in TCGA cancer in terms of the annotation mapping file provided by UCSC Xena.**
(TIF)

**S9 Fig. Correlation between *P*-values (in–log10 scale) of the methylation-expression association for all genes with meditation effects with various distances before the transcription start site (TSS) so that the promoter can be included.** Here, various distances before the TSS were extended, ranging from 500bp to 5000bp with an increment of 500bp. For each extension, the methylations within that extended region and gene body were included to examine their relationship with gene expression using the variance-component score test. The *P*-values calculated with methylations within a 500bp upstream of the TSS were treated as the reference.
(TIF)

**S1 Table. Estimated and true proportion parameters (mean and standard deviation) in the three-component mixture null distribution under the five simulation scenarios and different sample sizes and different numbers of mediators.**
(DOCX)

**S2 Table. Comparison of the effect of different numbers of mediators on the bias of the estimators for these proportion parameters.**
(DOCX)

**S3 Table. Data process of the ten TCGA cancers used in our mediation analysis.**
(DOCX)

**S4 Table. Basic characteristics of the ten TCGA cancer datasets.**
(DOCX)

**S5 Table. Results for testing for the proportional hazards assumption in the used Cox model.**
(DOCX)

**S1 Text. Effects in survival mediation analysis with multiple exposures, and identifiability assumptions.**
(DOCX)

**S2 Text. Intersection-union test.**
(DOCX)

**S3 Text. Three-component mixture null distribution.**
(DOCX)

**S4 Text. Estimation of proportion parameters.**
(DOCX)

## Acknowledgments

We would like to express our gratitude to TCGA for making genomic datasets publicly available and are indebted to all the investigators and participants contributed to this project. Data analyses and simulations in the present study were carried out with the high-performance computing cluster that was supported by the special central finance project of local universities for Xuzhou Medical University.

## Author Contributions

**Conceptualization:** Ping Zeng.

**Data curation:** Zhonghe Shao, Ting Wang.

**Formal analysis:** Zhonghe Shao, Ting Wang.

**Funding acquisition:** Ping Zeng.

**Methodology:** Zhonghe Shao.

**Project administration:** Ping Zeng.

**Resources:** Zhonghe Shao.

**Software:** Zhonghe Shao.

**Supervision:** Shuiping Huang.

**Validation:** Meng Zhang, Zhou Jiang.

**Visualization:** Zhonghe Shao, Meng Zhang.

**Writing – original draft:** Zhonghe Shao.

**Writing – review & editing:** Zhonghe Shao, Ping Zeng.

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
