## [Decision Letter · Decision Letter 0]

16 Mar 2021

Dear Dr. Zeng,

Thank you very much for submitting your manuscript "IUSMMT: survival mediation analysis of gene expression with multiple DNA methylation exposures and its application to cancers of TCGA" for consideration at PLOS Computational Biology.

As with all papers reviewed by the journal, your manuscript was reviewed by members of the editorial board and by several independent reviewers. In light of the reviews (below this email), we would like to invite the resubmission of a significantly-revised version that takes into account the reviewers' comments.

We cannot make any decision about publication until we have seen the revised manuscript and your response to the reviewers' comments. Your revised manuscript is also likely to be sent to reviewers for further evaluation.

Sincerely,

Oscar Rueda

Guest Editor

PLOS Computational Biology

Florian Markowetz

Deputy Editor

PLOS Computational Biology

Reviewer's Responses to Questions

**Comments to the Authors:**

Reviewer #1: Review is uploaded as a .docx attachment.

Reviewer #2: The authors claim that the found genes are causal. This is highly unlikely and should be noted. The authors should likely motivate these as informative tools for highlighting regions of interest as opposed to real causal signals.

In addition, the required assumptions for a causal interpretation are not mentioned til the method (referencing to S1 text line 479), these should likely be mentioned much earlier in the paper perhaps in the introduction.

If the proportional hazards assumption was violated, could standard approaches be used to correct for this?

How were DNA methylation markers mapped to genes? Was it solely based on distance? If so what was the threshold?

How was DNA methylation and gene expression standardized?

Is the data already QCed to adjust for potential batch or technical effects? The authors mention they restrict to those people of European Ancestry, was this verified through self-report or SNP data? The number of potential mediation genes seems like there is likely some inflation or unmeasured confounder in the analysis. The supplement says quality control but no further details are given.

Was cell type composition of the DNA methylation adjusted for? If not, it should be as it could be driving the associations.

In SFig 6 a ch DNA methylation marker is listed. Were non-cg DNAm probes left in the analysis? These are typically removed.

In the DNA methylation data, were DNAm sites that map to the sex chromosomes removed? Were cross reactive probes removed (PMCID: PMC3592906).

In the supplement materials discussing the \\lambda values it reads as if it can be set above 1, “It needs to highlight that a value of λα or λβ that is much closer to one can lead to a higher confidence to guarantee the null of or . However, a larger λα or λβ”. Larger than 1? Or just lambda values close to 1.

What is the practical guidance for researchers for \\lambda? Setting equal to 0.5?

In the supplement figures, the authors typically show the qq-plot and then show the histogram plots for the real data application. Would be best to be consistent. Perhaps showing the qq-plot for Supplement Figure 5?

How many genes are involved in Table 1

For the power in Figure 3: How many genes was this examined over? Is the power defined as the number of times the 1% of potential genes was selected at the desired TIE?

For the 6 different simulation scenarios: Is there a better way to display this information. At the current moment feels like a bit much and hard to take in.

Can the authors provide some intuition on why the estimation of the composite components is so poor under the dense null case

The authors discuss the power (S1 Fig) prior to the TIE (S2 Fig). While S1 appears motivated by the estimation of the parameters perhaps there is a way to discuss the TIE prior

The discussion of \\gamma^{total} seems unnecessary and distracts from the focus of the analysis, the NIE. Why the decision to include?

Is there any biological reasoning for why there are so many signals for KIRP and survival compared to the other cancers?

Minor

Line 175: typo “note”, think meant not?

Line 357: DNA methylation impacting gene expression is not a novel result

Figure 1: Perhaps better to have two DAGs? Reads almost like \\gamma^{total} is part of the same mechanism.

“Methylations”-> This is not typically how it is referred to as. It’s usually DNA methylation at a CpG site or methylation at probe.The authors claim that the found genes are causal. This is highly unlikely and should be noted. The authors should likely motivate these as informative tools for highlighting regions of interest as opposed to real causal signals.

Reviewer #3: Uploaded as an attachment.

**Have all data underlying the figures and results presented in the manuscript been provided?**

Reviewer #1: **No: **Data underlying tables from simulation studies and publically available TCGA data was not available for review.

In the absence of providing the underlying data, depending on PLOS Computational Biology policy, could the code be included to generate the simulated datasets/ analysis, and also code that follows the application in TCGA from the publically available data to the plots/ tables?

Reviewer #2: Yes

Reviewer #3: Yes

PLOS authors have the option to publish the peer review history of their article (what does this mean?). If published, this will include your full peer review and any attached files.

Reviewer #1: **Yes: **Rajbir Nath Batra

Reviewer #2: No

Reviewer #3: **Yes: **Majed Mohamed Magzoub
---

## [Decision Letter · Decision Letter 1]

6 Jul 2021

Dear Dr. Zeng,

We are pleased to inform you that your manuscript 'IUSMMT: survival mediation analysis of gene expression with multiple DNA methylation exposures and its application to cancers of TCGA' has been provisionally accepted for publication in PLOS Computational Biology.

Best regards,

Oscar Rueda

Guest Editor

PLOS Computational Biology

Florian Markowetz

Deputy Editor

PLOS Computational Biology

Note from the editor: I would try to address the minor comments made by the authors

Reviewer's Responses to Questions

**Comments to the Authors:**

Reviewer #1: The authors have provided a revised manuscript regarding their novel gene-centric approach called IUSMMT (Intersection-Union Survival Mixture-adjusted Mediation Test) to delineate causal genes that mediate the effect of DNA methylation on cancer-specific survival. The authors responded to each of the specific comments raised by the reviewers and revised the manuscript accordingly.

In response to the reviewer’s comment regarding distinguishing passenger methylation events from mediation methylation events, the authors have performed an ad-hoc reverse analysis by regressing DNA methylations on expression for all the identified mediating genes based on a multivariate model. Could the authors include statistical details (with equations) of this additional analysis in the Methods section, as they have for the rest of the IUSMMT construction?

The authors should ensure that the IUSMMT method is released as a publicly available package to enable application to other datasets. The computational code for the analysis should also be made readily available to the research community.

My overall impression remains that the manuscript is a matter of interest since the new approach (IUSMMT) has potential utility in understanding the mechanism underlying the regulatory role of DNA methylation in cancer progression; and can aid in molecular classification as well as generate new insights into potential therapeutic targets. This paper will be a good addition to PLOS Computational Biology.

Reviewer #3: The authors have adequately addressed the necessary changes suggested in previous major and minor concerns. The manuscripts quality and content merit publication.

Reviewer #4: The authors proposed a novel method IUSMMT to link DNA methylation and survival via gene expression.

The model and results looks pretty interesting.

I am pleased to read the revision along with responses to all review comments. And I think the authors have addressed most of the concerns, and the current version is good to be accepted.

However, I do have one major question and one minor comment.

Major question: Per reviewer 1's Q3, the authors replied "However, these identified genes cannot include proliferation or cell cycle related genes. Following the suggestion, we now clearly pointed out this issue in lines 459-462 on page 22." I am super confused by what the sentence means here. Why cannot the identified genes include proliferation or cell cycle related genes? The statement does not make sense to me. Reading the version, I guess the authors meant that they are not sure whether the identified genes are cell cycle related genes, right? But the author could do a simple enrichment analysis on these genes using the GO cell cycle gene list.

I also have a minor comment related to reviewer 2's major comment 6.

Although I don't think the authors must consider cell type composition in their model as they have already done some extensive job, they should at least mention two points in discussion: 1) the DNAm levels are affected by cell type composition - cite Jaffe AE and Irizarry RA 2014 Genome Biology; 2) the cell type composition is an important factor for survival - see Figure 4 (Ratio of infiltrating PMNs to plasma cells is prognostic in diverse solid tumors) in Gentles AJ et al 2015 Nature Medicine. The lack of consideration in cell type composition is indeed the weak point of the current model and should be explored in future work.

**Have the authors made all data and (if applicable) computational code underlying the findings in their manuscript fully available?**

Reviewer #1: **No: **

Reviewer #3: Yes

Reviewer #4: Yes

PLOS authors have the option to publish the peer review history of their article (what does this mean?). If published, this will include your full peer review and any attached files.

Reviewer #1: **Yes: **Rajbir Nath Batra

Reviewer #3: **Yes: **Majed Mohamed Magzoub

Reviewer #4: No

---

## [Editor Report · Acceptance letter]

25 Aug 2021

PCOMPBIOL-D-21-00050R1 

IUSMMT: survival mediation analysis of gene expression with multiple DNA methylation exposures and its application to cancers of TCGA

Dear Dr Zeng,

I am pleased to inform you that your manuscript has been formally accepted for publication in PLOS Computational Biology. Your manuscript is now with our production department and you will be notified of the publication date in due course.

With kind regards,

Livia Horvath
